# Promises and Challenges of Cell-Based Therapies to Promote Lung Regeneration in Idiopathic Pulmonary Fibrosis

**DOI:** 10.3390/cells11162595

**Published:** 2022-08-20

**Authors:** Alejandro Egea-Zorrilla, Laura Vera, Borja Saez, Ana Pardo-Saganta

**Affiliations:** 1Institute for Lung Health (ILH), Department of Internal Medicine, Justus-Liebig University, Universities of Giessen and Marburg Lung Center (UGMLC), German Center for Lung Research (DZL), 35392 Giessen, Germany; 2Cardio-Pulmonary Institute (CPI), Department of Internal Medicine, German Center for Lung Research (DZL), Justus Liebig University, 35392 Giessen, Germany; 3Solid Tumors Program, Division of Oncology, CIMA Universidad de Navarra, 31008 Pamplona, Spain; 4Department of Hematology-Oncology, CIMA Universidad de Navarra, 31008 Pamplona, Spain

**Keywords:** pulmonary fibrosis, tissue homeostasis, lung regeneration, stem/progenitor cells, alveolar epithelial cells, cell therapy

## Abstract

The lung epithelium is constantly exposed to harmful agents present in the air that we breathe making it highly susceptible to damage. However, in instances of injury to the lung, it exhibits a remarkable capacity to regenerate injured tissue thanks to the presence of distinct stem and progenitor cell populations along the airway and alveolar epithelium. Mechanisms of repair are affected in chronic lung diseases such as idiopathic pulmonary fibrosis (IPF), a progressive life-threatening disorder characterized by the loss of alveolar structures, wherein excessive deposition of extracellular matrix components cause the distortion of tissue architecture that limits lung function and impairs tissue repair. Here, we review the most recent findings of a study of epithelial cells with progenitor behavior that contribute to tissue repair as well as the mechanisms involved in mouse and human lung regeneration. In addition, we describe therapeutic strategies to promote or induce lung regeneration and the cell-based strategies tested in clinical trials for the treatment of IPF. Finally, we discuss the challenges, concerns and limitations of applying these therapies of cell transplantation in IPF patients. Further research is still required to develop successful strategies focused on cell-based therapies to promote lung regeneration to restore lung architecture and function.

## 1. Introduction

Interstitial lung diseases (ILDs) represent a group of more than 200 diverse parenchymal pulmonary disorders that share similar clinical, radiographic, physiologic or pathologic manifestations. ILDs are mainly characterized by alveolar and interstitial inflammation and/or fibrosis, usually leading to decreased lung function that may eventually be fatal [1,2]. Idiopathic Pulmonary Fibrosis (IPF) is a chronic, progressive and fatal disease of unknown etiology and it is considered one of the most representative types of lung fibrosis within the ILDs [3]. Of note, in relation to its unknown etiology and its similarities with other ILDs, IPF diagnosis usually relies on the exclusion of alternative diagnoses and the identification of a pattern of progressive usual interstitial pneumonia (UIP) [4]. The radiologic pattern of UIP consists of bilateral reticulation and honeycombing specifically in the periphery and in the lower lobes, with a correspondent histologic UIP pattern consisting of heterogeneous paraseptal fibrosis with architectural distortion associated with the typical presence of fibroblast foci [2]. IPF usually occurs in older adults (68 years is the mean age of diagnosis) and its prevalence, incidence and mortality is higher not only in the aged population but also among males [1,3,5,6,7]. Furthermore, with an incidence of 0.09–1.30 per 10,000 people, an estimated prevalence of 0.33–4.51 per 10,000 people and a median survival of 3–5 years following diagnosis, IPF is regarded as a rare disease that is more lethal than most cancers [5,6].

IPF is characterized by the irreversible scarring of the distal lungs as a consequence of excessive accumulation of the extracellular matrix (ECM), rendering the lung stiff and compromising its normal function of gas exchange. This is reflected in a progressive decline in lung function and quality of life in most patients that eventually leads to respiratory failure and death [7,8]. Several therapeutic interventions have been tested to treat IPF, but none have been proven to succeed. Since inflammation was initially considered to be the key driver of IPF, corticosteroids and immunomodulators were recommended as a standard treatment by the 2000 ATS/ERS statement, but it was later discouraged by the PANTHER-IPF study [9]. Furthermore, the use of anticoagulation drugs, suggested due to the presence of a characteristic “pro-thrombotic state” in IPF patients, as well as the treatment with different monoclonal antibodies targeting specific proteins have not shown the desired efficacy [9]. Currently, only two oral antifibrotic drugs—nintedanib and pirfenidone—have been approved by the FDA for the treatment of IPF [10]. However, neither of them is able to completely halt the disease’s progression nor cure the disease. Lung transplantation is the only alternative in end-stage lung disease and it can prolong survival in the IPF patients that received a lung transplant [11]. Nevertheless, the old age and health status of the patients usually makes them ineligible for transplantation. 

Despite significant progress in our understanding of pulmonary fibrosis, the lack of an efficient treatment is a reflection of the limited insight into the pathophysiological mechanisms underlying the initiation and progression of this fatal disease. In fact, as stated above, IPF is still best described by its histopathological pattern of UIP, characterized by the loss of epithelial structures, interstitial collagenized fibrosis, microscopic honeycombing and presence of “fibroblast foci” with overlying hyperplastic epithelial cells [7,8]. The biological processes involved in the pathogenesis of lung fibrosis seem to be similar to those which regulate physiological wound healing, albeit they are dysregulated and exacerbated [12,13]. Physiological wound healing includes the resolution of the fibrotic scarring process once the source causing the injury disappears. This damage may be caused by pathogens, such as respiratory viruses or bacteria, dysbiosis or from environmental agents such as cigarette smoke or fumes [14,15,16,17]. However, when the damage is chronic or repetitive, tissue repair turns into an aberrant wound-healing response and pathological fibrosis takes place [18,19,20]. Although the etiology of IPF remains unknown, multiple lines of evidence point towards alveolar epithelial cell death and/or dysfunction, and the resulting basement membrane denudation and loss of epithelial integrity as the major trigger for the initiation and/or progression of lung fibrosis [21]. Yet, multiple cell types and events are implicated in IPF pathogenesis. Importantly, the disruption of the complex crosstalk between the alveolar epithelium and the neighboring mesenchyme may be crucial to driving fibrogenesis [2,7,12]. Moreover, the close relationship between the alveolar epithelium and endothelium and their ability to transition to mesenchymal-type cells highlights the importance of keeping an integrative view that pays attention to the interaction between these cell types and their relations with the fibrotic phenotype [4]. Also, since the ECM is aberrantly accumulated in IPF, it is clear that the study of its contribution to IPF onset and progression is critical. Furthermore, the study of cellular senescence is of major relevance in IPF research, as well as other age-related factors including mutations affecting telomere maintenance, since IPF shows most of the hallmarks of aging and is thought to result as a consequence of accelerated aging [2,18]. Understanding the intricate environment of the fibrotic lung will help us to find successful strategies to block the progression of pulmonary fibrosis. However, in this context, mechanisms of repair are inefficient. Thus, further investigation about the cell types with regenerative capacities to restore functional lung tissue is key in order to develop therapies that promote or induce lung regeneration. In this review, we aim to summarize the most recent findings in the field, from cellular mechanisms implicated in lung regeneration to the latest cell-based therapies for the treatment of IPF.

### 1.1. The Alveolar Compartment and the Development of Pathological Fibrosis

Alveoli are cup-shaped cavities in the distal lung where gas exchange takes place. The alveolar epithelium is mainly composed of two cell types, alveolar type 1 (AT1) and alveolar type 2 (AT2) cells. On the one hand, AT1 cells cover 95% of the alveolar epithelium, have a squamous shape and are in close relation with endothelial cells of the capillary plexus, forming the gas-exchange surface of the lung. On the other hand, AT2 cells are cuboidal surfactant-producing cells (pulmonary surfactant reduces surface tension preventing the lungs from collapsing upon every breath) and serve as alveolar stem cells in the adult lung [22]. Despite these cells being the main cell lineages that form the general architecture of the alveoli, there are other epithelial and endothelial cell subtypes and a complex mixture of both immune and mesenchymal cells that are present in the interstitial space of the lung between the alveolar units [23,24,25,26]. In this regard, single cell RNA-sequencing (scRNAseq) studies are starting to provide enough information to create an accurate census of the cells present in the murine and the human lung in health and disease. The human lung is comprised of more than 50 transcriptionally distinct cell populations whose distribution varies across the human respiratory tree [25]. Furthermore, the advent of spatial-omics is also shedding light onto the spatial distribution of the different cell populations within the lung [27]. The use of these technologies will help to overcome the difficulties in the extrapolation of the results of studies into the human lung brought by the inherent differences between the mouse and human lungs including their size and cellular composition.

After alveolar epithelial cell injury, a wound-healing response is activated to restore the damaged tissue (Figure 1). Remaining epithelial cells with the capacity to regenerate the alveolar epithelium are activated (proliferate and/or differentiate into AT1 and AT2 cells) to replace lost cells, restoring barrier integrity and lung function [28]. However, upon repetitive damage, the constant death of alveolar epithelial cells and the inability of progenitor cells to repopulate the alveoli results in an aggravated and persistent inflammatory response with the release of several cytokines including a transforming growth factor (TGF)β1 that has been shown to play a crucial role in the development of fibrosis [29]. This inflammatory response ultimately elicits the recruitment and activation of fibroblasts. These fibroblasts proliferate, migrate and differentiate into myofibroblasts, acquiring a contractile phenotype similar to that of smooth muscle cells, showing apoptosis resistance and producing exaggerated amounts of collagen and other ECM proteins [30,31]. This leads to the formation of fibroblast foci, regions of highly proliferative myofibroblast accumulation that are located immediately adjacent to regions of hyperplastic or apoptotic epithelial cells, which grow over time leading to a massive accumulation of the ECM that manifests as the thickening of the interstitium of the alveolar walls and tissue stiffening [31,32]. As a result, gas exchange is hampered, which may lead to respiratory failure and death in a worst-case scenario. 

### 1.2. Modeling Lung Fibrosis

To model lung fibrosis in vivo, different mouse injury models that resemble different aspect of the pathophysiology of the fibrotic process have been developed (Table 1). The bleomycin-induced pulmonary fibrosis model in mice is probably the most broadly used model to study specific processes involved in the lung fibrogenesis. However, this model has several limitations since a single intratracheal injection of bleomycin gives rise to an inflammatory-mediated fibrotic response that resolves naturally [33]. Furthermore, similar in vivo models have been used to study lung fibrosis [33,34,35] including silica and asbestos exposure [36,37], hyperoxia [38], acid instillation [39], cytokine overexpressing [40], fluorescent isothiocyanate [41], radiation exposure [42], and familial models [43] (Table 1). However, none of these models mimic the natural progression seen in patients with IPF where fibrosis is irreversible and tends to increase with time, but have allowed researchers to understand specific aspects of the disease [44]. In the last few years, a chronic model of repetitive administration of bleomycin that results in irreversible fibrosis and lung function decline has been optimized [45,46]. This model will serve as a better setting to evaluate lung fibrosis emulating major hallmarks of human IPF.

On the other hand, there are groundbreaking approaches that use human primary cells and tissues from healthy donors or IPF patients to study lung fibrosis. Some of these models are based on 2D or 3D culture techniques such as hydrogels, precision cut lung slices (PCLS), lung organoids and lung-on-chip [58,59,60,61,62,63]. Novel ex vivo models have been developed to evaluate early fibrosis and to study tissue repair. For example, the administration of a cocktail of profibrotic factors including TGFβ, TNFα, PDGF-AB and LPA on human-derived PCLS has been demonstrated to induce early fibrotic changes facilitating the study of early-stage IPF pathogenesis [58]. Another model based on the exposure of a restricted area of a PCLS to an acid injury (the Acid Injury and Repair model) revealed the cellular responses in injured vs. uninjured regions resembling the heterogeneous pattern observed in lung disease and suggesting that this model may be helpful in understanding lung disease pathogenesis and tissue repair [59]. In addition, organoids derived from human pluripotent stem cells have been proved to be a reliable tool to model fibrotic lung disease [60]. In general, while 3D models recapitulate better the real pathophysiological process of lung scarring does, the availability of human lung tissue remains scarce when compared to that of mice, and the interindividual variability makes it difficult to establish robust preclinical disease models [35]. Furthermore, each model has its own benefits and drawbacks. Hydrogels are ideal for conducting cell-specific therapeutic strategies and rather useful for the study of fibroblasts in the context of fibrosis [63]. However, they are not suitable for the study of cellular interactions [64]. In the case of PCLS, although they contain nearly all cell types present in the lung, preserve lung architecture and are metabolically active in the culture, explants last only up to 14 days in culture or up to 21 days with the support of hydrogel encapsulation [65]. Lung organoids are a good tool to study mesenchymal-epithelial crosstalk and provide the opportunity to perform high throughput analyses, nonetheless, the lack of the vasculature and the immune compartment poses a great bias in the results obtained through this model [64]. On the contrary, while lung-on-chip models allow the integration of vascular and immune cells, the experimental throughput is low [64]. Thus, the development of better human models is of high importance to understand IPF and find successful treatments for it.

## 2. Regeneration and Stem/Progenitor Cells

### 2.1. Epithelial Stem and Progenitor Cells Are Contributing to Regenerate the Alveolar Epithelium

The adult lung epithelium is highly susceptible to damage since, due to its major function of gas exchange, it is constantly exposed to insults from the external environment including toxic chemicals and pathogens—virus or bacteria—which makes the lung vulnerable. However, although the lung is relatively quiescent at homeostasis, it shows a significant regenerative capacity in response to injury. This remarkable ability to repair damaged tissue is due to the presence of multiple stem and progenitor cell populations that quickly respond to injury (Figure 2) [66,67,68,69,70,71].

In recent years, the advent of technologies such as single cell RNA sequencing (scRNASeq), spatial transcriptomics and advanced imaging has allowed to identify novel cell types of the lung and subpopulations, as well as transcriptional patterns and dynamics in healthy and diseased tissues. Multiple studies using single cell transcriptomics in combination with lineage tracing, lung injury models and organoid cultures have revealed that depending on the site of the damage (airway vs. alveoli) as well as the type and the extent of injury, different epithelial progenitor populations with the capacity to self-renew and differentiate into the epithelial lineages of the tissue respond to restore tissue homeostasis [67,69,72,73].

In the alveolar compartment, lineage tracing experiments have demonstrated that AT2 cells serve as progenitor cells of the homeostatic adult alveolar epithelium that are able to proliferate and differentiate into AT1 cells [74,75,76,77,78,79]. AT2 cells are mostly quiescent in the steady-state lung, but during repair after influenza infection, bleomycin exposure, hyperoxia, pneumonectomy-induced injury and genetic ablation of alveolar epithelial cells, AT2 cells quickly re-enter the cell cycle and produce new alveolar epithelial cells to keep the integrity of the alveolar epithelium and repair damage [66,71,78,79,80,81,82,83,84,85].

Although initially AT2 cells were thought to be a homogeneous population, recent studies have identified different subpopulations with higher regenerative capacities [86,87,88]. A subset of AT2 cells characterized by the expression of Axin2, a transcriptional target of Wnt signaling, and termed alveolar epithelial progenitors (AEPs) [88] are found at sites of severe damage and serve as the principal progenitor cell population during injury-induced alveolar regeneration [87,88]. Axin2+ AT2 cells were identified by two independent studies, both of them showing that these cells are highly proliferative and can further differentiate into AT1 cells following bleomycin- and influenza virus-induced lung injury [71,87,88]. The proportion of Axin2-expressing AT2 cells significantly varied between these two studies likely due to the use of different experimental strategies, but both demonstrated the expansion of this subpopulation after injury and their contribution to alveolar repair [87,88]. Another subset of AT2 cells is characterized by the expression of CD44 and exhibit high rates of proliferation suggesting a role in alveolar regeneration [86]. Whether Axin2+ and CD44+ AT2 cells are the same or a different subpopulation is still to be confirmed.

In addition to AT2 cells, considered as primary contributors to post-injury alveolar regeneration, various epithelial progenitor populations have been shown to participate in regeneration after damage [77,78,89].

BASCs (Bronchoalveolar Stem Cells) are found at the bronchoalveolar duct junction and they have been long understood to be stem cells that give rise to both AT2 cells and Club cells [90,91]. They were first defined by their anatomical location and the co-expression of the club cell marker, SCGB1A1, and the AT2 cell marker, SFTPC, suggesting their potential to differentiate into both cell types under a condition of lung injury. It was not until recently that it has been demonstrated, using two different approaches to lineage trace these cells in vivo, that indeed BASCs serve as progenitors of the distal airway and alveolar epithelium, being able to generate club cells, ciliated cells, AT2 and AT1 cells, confirming their contribution to lung regeneration [92,93,94].

Besides BASCs, airway epithelial stem/progenitor cell populations have been shown to contribute to both airway and alveolar regeneration [89,90,91,93,95,96,97,98,99,100,101,102,103,104,105,106,107]. Among these populations, several subsets of club cells in the terminal bronchioles, including Upk3a+ variant club cells, H2-K1 high club cells and MHC-II+ club cells, have been identified to give rise to AT2 cells in different models of severe damage [83,95,97,98,104].

A rare population of p63-expressing cells was identified in terminal bronchioles and their contribution to airway and alveolar regeneration was examined [99,101,102,105]. After severe injury such as bleomycin exposure or influenza infection, this rare population of the airway (Lineage Negative Epithelial Progenitors (LNEPs) [101]; Distal Airway Stem Cells (DASCs) [99]) activate the expression of Krt5, expand and migrate to damaged regions of the alveoli. However, these cells rarely give rise to AT2 cells nor to AT1 cells [101,105], but instead form pod-like metaplastic structures in which the alveoli are not repaired which persist for a long time and are likely thought to fill the injured space to preserve structural integrity. Nevertheless, Xi et al. demonstrated using stringent lineage tracing tools that hypoxia and Notch and Wnt signaling regulate the fate decision of p63+ progenitors to either regenerate the alveolar epithelium or to form these pod-like structures [89]. 

Surprisingly, recent evidence demonstrates that even terminally differentiated AT1 cells are able to dedifferentiate and give rise to AT2 cells after pneumonectomy and hyperoxia-induced lung injury [108,109], reflecting the significant plasticity of lung cells and conferring the lung as a valuable mechanism of repair. A tonic activation of YAP/TAZ is required to maintain AT1 identity, and their loss induces their differentiation into AT2 cells [109]. This is of high interest since it reveals mechanisms of the active maintenance of quiescence [67] necessary to keep tissue homeostasis similar to the Notch2 tonic activation in Club cells [110] that has been previously reported and that is required to maintain their cell identity and preserve adequate proportions of each cell type in the airway epithelium. 

### 2.2. Mechanisms of Alveolar Regeneration 

Multiple signaling pathways including BMP, FGF, Notch pathway, Wnt signaling, TGFβ, Hippo pathway, NF-κB and mechanotransduction pathways have been described to regulate AT2 proliferation and/or differentiation [82,85,100,111,112,113,114,115,116,117,118,119]. These signals are provided by the microenvironment, comprised of different cell types including PDGFRα lipofibroblasts, pulmonary endothelial cells and alveolar immune cells, and the extracellular matrix [71,117,120,121] which is known to influence stem/progenitor cell behavior in many different tissues including the lung [122]. Thus, understanding the communication between AT2 cells and the different cell types in their niche is crucial to unravel the mechanisms involved in lung regeneration during normal physiological repair as well as providing insight into maladaptive repair and pathological processes.

Importantly, the precise mechanism regulating the differentiation of AT2 cells into AT1 cells has been recently revealed. Interestingly, the stepwise pathway that an AT2 cell goes through in their way to become an AT1 cell includes the transition through an intermediate state required to achieve successful regeneration. Using a combination of scRNASeq, RNA velocity, lineage tracing, distinct injury models and organoid cultures, three independent groups identified intermediate cells—“pre-alveolar type-1 transitional cell state” (PATS) [123], Krt8+ alveolar differentiation intermediate (ADI) cells [107] and damage-associated transition progenitors (DATP) [124]—characterized by the expression of Cldn4 and Krt8 [107,123,124] in addition to Krt19 and Sfn in PATS [123], and these are rarely found in the homeostatic lung but arise after injury. These cells were found in organoids generated by AT2 cells and in a variety of lung injury models including LPS [123], bleomycin administration [107,123,124], genetic AT1 cell ablation [123], pneumonectomy [123], adult hyperoxia and neonatal hypoxia and hyperoxia with Influenza type-A infection [107], suggesting that it is a common injury response. Lineage tracing experiments demonstrated that these intermediate cells derive from AT2 cells [107,123,124] with a contribution of airway MHC-II+ club cells that give rise to alveolar lineages through Krt8+ ADI cells [124], likely depending on the site and the severity of injury.

One of the most striking observations during this process is the activation of pathways associated with stress and this may be the reflect of the significant change in morphology (cell shape and structure) that AT2 cells (cuboidal) need to accomplish in order to differentiate into AT1 cells (flat and thin) [123]. Notably, these transitional cells show a unique transcriptional signature characterized by the activation of pathways of DNA damage such as p53 signaling, cellular senescence and TGFβ signaling [85,107,123,124,125,126,127]. Interestingly, the pharmacological activation of p53 induced AT2 differentiation, while the genetic inactivation of p53 hampered cells at the transitional stage, demonstrating that p53 signaling is sufficient and necessary to promote AT2 differentiation through PATS to give rise to AT1 cells after injury [123]. Importantly, these studies reveal that normal alveolar regeneration involves a transitional state that exhibits features of senescence which usually occurs in aged cells and is thought to be irreversible. Thus, these observations support that reversible physiological senescence may be a program of normal tissue regeneration and maintenance vs. pathological senescence occurring in disease. 

Previous studies had also observed intermediate states between AT2 and AT1 cells. The accumulation of an AT2 transitional state following pneumonectomy was observed in Cdc42-null mutants, demonstrating that elevated mechanical tension arrested regeneration at an intermediate state and resulted in fibrosis [85]. Similarly, the inactivation of Dlk1, a suppressor of Notch signaling, led to the accumulation of an intermediate cell population during AT2 to AT1 cell transition triggered by a Pseudomonas aeruginosa infection, suggesting that Notch activation is needed to initiate alveolar regeneration [119]. Although it is unknown whether these cells are similar or distinct from PATS, ADI or DATP cells, altogether these studies confirm that an intermediate AT2-AT1 cell state exists during alveolar regeneration.

As previously mentioned, the microenvironment is known to influence AT2 behavior [71] and critical cues from their niche drive the initiation of AT2 differentiation. The analysis of the interactome during alveolar regeneration revealed that intermediate cells display a distinct receptor-ligand connectome with mesenchyme and macrophages [107]. In fact, this process of AT2 differentiation has been shown to be regulated at least in part by inflammatory cytokines, in particular IL-1β which is produced by interstitial macrophages [124], mechanical forces [85] and Notch signaling [119]. Importantly, the temporal regulation of these signals during AT2-AT1 transition is crucial, since although Notch activation and IL-1β are necessary to initiate AT2 differentiation, their persistence impedes terminal differentiation [119,124]. Further investigation will reveal other regulatory networks involved in this process, whether other subsets of “regenerative” AT2 cells previously unrecognized such as Il1r1+ AT2 cells [124] exist, or how is this differentiation process altered in aged AT2 cells.

### 2.3. Alveolar Regeneration in Human Lungs

Most of our knowledge of lung regeneration derives from studies performed using mouse models. However, important differences exist between the mouse and the human respiratory system. Probably one of the most important differences is the lack of respiratory bronchioles in the mouse lung, which seems to be a key site for the development of diseases. Thus, big efforts are currently being made in order to extend the previous findings into studies of the pathogenesis of human lung diseases such as IPF [107,123].

Although human alveolar regeneration is poorly characterized, recent studies have identified novel epithelial progenitors with the capacity to regenerate the human alveolar epithelium. Kadur Lakshminarasimha Murthy et al. identified a novel bi-potential cell population that they called AT0 that can normally be found in the alveolar sacs and that emerges and expand after injury [128]. Using a non-human primate model of lung injury together with human organoids and tissue samples, they found evidence that during regeneration, AT2 cells transiently go through an AT0 cell state in their differentiation into AT1 cells or secretory cells of the terminal respiratory bronchioles (TRB-SCs). Of note, this process is different than the one observed in the mouse lung and that involves a different transient state known as PATS [123].

On the other hand, Basil et al. identified a unique airway secretory cell population in human respiratory bronchioles that they called RAS cells, with the capacity to differentiate into AT2 cells in a process regulated by Notch and Wnt signaling [129]. RAS cells are characterized by the expression of SCGB3A2 and lack the expression of the club cell marker, SCGB1A1, and the AT2 cell marker, SFTPC, representing a transcriptionally intermediate state between canonical airway secretory cells and AT2 cells. This novel epithelial progenitor serves as a putative progenitor for the AT2 cell lineage, and the RAS cell-AT2 cell rapid and unidirectional differentiation has been found to be altered in disease such as COPD [129].

In chronic lung diseases, mechanisms of repair are dysregulated and inefficient [72]. Interestingly, an accumulation of SCGB3A2+ AT2 cells was observed in humans and ferrets that were exposed to cigarette smoke and in COPD lungs [129]. Similarly, increased numbers of AT0 cells were found in lungs of patients with COPD and with IPF [128]. The accumulation of these transitioning cells could represent a more active attempt to regenerate the damaged tissue, or a blockade in regeneration. The fact that AT2 cells in IPF lungs persistently express markers of intermediate cells [107,123] suggests that AT2 cells in fibrotic lungs do not have the capacity to further differentiate into AT1 cells [85,126]. Of note, these intermediate cells also exhibit a profibrogenic signature [107], maybe contributing to amplify the fibrotic loop. Importantly, PATS, ADI and DATP signatures were also found in the fibrotic regions of lungs from IPF patients [107,124,128]. Yet, these cells would be more similar to Krt17+ p63+ Krt5- basaloid cells found in sites of active fibrosis, predominantly accumulated at the edge of fibroblast foci in IPF lungs [107,123,130,131,132,133]. These observations suggest that the depletion of these transitioning cells or the promotion of their terminal differentiation into AT1 cells could represent previously unexplored therapeutic avenues. 

Understanding the tissue-resident stem/progenitor cells involved in lung repair and regeneration post-injury as well as their interactions with the environment and the mechanisms that regulate reparative responses is crucial to developing strategies to promote regeneration and restore functional tissue in chronic lung disease together with therapies that halt the progression of the disease. This knowledge will facilitate the development of new therapeutic approaches for treating respiratory diseases.

Strategies for therapeutic lung regeneration are actively pursued and they include: (1) tissue engineering new lungs in vitro [134], (2) pharmacological manipulation of endogenous healthy cells to induce repair [67], and (3) administration of exogenous stem or progenitor cells to damaged lungs [135]. Lung cell transplantation has been achieved in mice following severe injury by influenza infection, by naphthalene injury and followed by systemic irradiation or following a low dose of bleomycin [101,102,136,137]. However, these strategies are in the early stages of development and questions about the successful engraftment of these cells, or the actual improvement of lung function remain. In order to apply this therapeutic approach in humans, many open questions need to be answered: What is the best route of cell administration? Is it directly through the airway or by IV injection? Will transplanted cells engraft and survive? How many cells are required? What will be the fate of these cells? Will they differentiate into alveolar epithelial cells in the environment of an injured lung, or will a proper microenvironment also be required to instruct adequate differentiation/regeneration? Which is their mechanism of action? Would these cells exert their effects by engraftment facilitating direct cell-cell contact or through paracrine secretion of growth factors, cytokines and hormones? The transplantation of healthy cells with a regenerative capacity that could contribute to tissue repair is described in further detail below. 

## 3. Cell Therapy in Lung Fibrosis

As stated above, pirfenidone and nintedanib are the only drugs approved for the treatment of IPF. Although these two drugs have shown efficacy in reducing the rate of decline in lung function and slowing the pace of the disease’s progression, they are not able to halt the disease’s progression [138,139]. In addition, both compounds have shown significant side effects such as gastrointestinal- and skin-related adverse events in the case of pirfenidone, and diarrhea in the case of nintedanib [138,139]. In order to cure pulmonary fibrosis, we must probably combine therapies addressed towards blocking pathogenesis and promoting tissue regeneration to completely restore lung tissue that is able to successfully function.

Stem cells and endogenous lung progenitor cells have been studied for many years as a therapy to promote tissue repair in chronic diseases because of their regenerative potential including their self-renewal capacity and ability to differentiate into different cell types of their tissue [140]. Remarkably, stem cell-based tissue engineering aims to mimic the native stem cell niche and maintain stem cell function within the graft by providing appropriate microenvironmental cues in a controlled and reproducible fashion to facilitate its application into human diseases including IPF [141].

In the last decade, cell therapy has been investigated for the treatment of IPF, including the use of a variety of cell types such as lung epithelial cells, specifically AT2 cells [142], induced pluripotent stem cells (iPSCs), and mesenchymal stem cells (MSCs) isolated from bone marrow stroma [143] and those from adipose tissue or from other tissues (Figure 3; Table 2). Notably, both endogenous alveolar epithelial and mesenchymal stem cells are the most widely investigated ones for the treatment of IPF.

### 3.1. Preclinical Mouse Studies

#### 3.1.1. Epithelial Cells: Alveolar Type 2 Cells

AT2 cells function as stem cells in the adult lung [78], maintaining tissue homeostasis and contributing to alveolar repair after damage [144]. However, in IPF, a significant portion of AT2 cells are lost or are dysfunctional and are replaced by fibroblasts and myofibroblasts [142]. In previous preclinical studies, Serrano-Mollar and colleagues revealed the therapeutic effect of the intratracheal transplantation of AT2 cells resulting in a reduction of the extent of experimental pulmonary fibrosis [145,146]. AT2 cell treatment at either 3, 7 or 15 post-bleomycin installation showed decreased fibroblast proliferation and prevented accumulation of the ECM [146]. Furthermore, the injection of AT2 cells 14 days post bleomycin-induced lung injury resulted in a restoration of the surfactant levels [145] and similarly, the administration of AT2 cells 7 days after injury improved the lung function based on elastance and compliance measurements [147]. Interestingly, Cores and colleagues established that adult lung spheroid cells (LSCs) are an intrinsic source of therapeutic lung stem cells [148]. The administration of these cells right after bleomycin instillation in a rat model decreased the extent of fibrosis, reduced apoptosis, protected alveolar structures and increased angiogenesis [148].

#### 3.1.2. Adult Mesenchymal Stromal/Stem Cells

Adult mesenchymal stromal/stem cells (MSCs) are multipotent cells with the ability to differentiate into a wide range of cell types [149,150]. Not only murine, but also human MSCs have been used in studies involving animal models [151,152]. According to several studies, MSCs can home to damaged tissues when systemically administered via intravenous (IV) or intraperitoneal (IP) injection [153,154]. Ortiz and colleagues reported for the first time that bone marrow MSC-derived cells (BM-MSCs) injected into the jugular vein immediately after bleomycin instillation were able to engraft to sites of lung injury, reducing both inflammation and collagen deposition [155]. In contrast, beneficial effects were not observed when they were given 7 days after the injury occurred [155]. Similarly, other studies also showed that only the very early administration of BM-MSC after bleomycin decreased the damage that incurred [156,157,158,159,160]. Moreover, the combination of BM-MSC with nintedanib in bleomycin-induced lung fibrosis in rats showed anti-inflammatory and antifibrotic activity [161]. Furthermore, tissue-resident MSCs have also been shown to protect lung integrity after bleomycin instillation in mice [162].

MSCs have been extensively used in a number of clinical trials to find treatment options for many different diseases. In fact, it has been shown that MSCs have potent anti-proliferative, anti-apoptotic, immune-modulatory and anti-inflammatory properties, besides their multilineage differentiation capacity, making them an attractive and promising option for diseases with no cure [149,163]. However, the majority of the preclinical studies administering BM-MSCs have used them as a preventive strategy rather than a treatment approach, and further research analyzing their effects after injury are required in order to validate their therapeutic potential [155,164,165]. The major limitation to evaluating the potential of BM-MSCs as a treatment for lung fibrosis is that these studies use the single-dose bleomycin model which induces reversible damage and fibrosis resolves naturally, narrowing the time window for these cells to exert their action. Thus, a model of established fibrosis like the chronic administration of bleomycin to resemble some of the major pathological features of human IPF [45,46] must be used to test their efficacy as a treatment. Yet, it is important to take into account that most of the studies were performed in rodents subjected to established injury models (i.e., bleomycin administration) that usually display a severe phenotype but do not reflect genetic or inter-strain variability and do not completely mimic the pathophysiology of the human disease [166,167]. 

Adipose mesenchymal stem cells (ADMSC) have been also evaluated [168,169,170,171,172]. The administration of adipose MSCs 7 days post-bleomycin treatment inhibited both a pulmonary inflammation and fibrosis which significantly improved the survival rate of mice in a dose-dependent manner [169]. In concordance, Lee and colleagues observed a reduction in the hyperplasia of epithelial cells and a reduction in the inflammation and fibrosis after the repeated ADMSC administration at 8, 10, 12 and 14 weeks in a chronic model of fibrosis [170]. Hence, ADMSCs could have positive effects even when they are administered once fibrosis has been established. A study using human derived ADMSC demonstrated that these cells significantly increase survivability and reduce organ weight and collagen deposition better than pirfenidone does after the intratracheal challenge with bleomycin in mice [172]. In general, ADMSCs have proven their efficacy in bleomycin-treated aged mice when they come from a young donor [171]. Interestingly, the age-dependent antifibrotic properties of these cells was demonstrated since old donor-ADMSC treatment in old bleomycin-treated mice did not reduce fibrosis and related markers [169]. Furthermore, another study used human placental mesenchymal stem cells of fetal origins (hfPMSCs) and this showed that inflammation and fibrosis in bleomycin-treated mice was alleviated after treatment with these cells [173]. Similarly, amniotic fluid stem cells (AFSC) have shown the potential to inhibit the development or progression of the fibrotic phenotype in both acute and chronic remodeling events in mice treated with bleomycin [174].

#### 3.1.3. Induced Pluripotent Stem Cells 

Several studies have transplanted pluripotent stem cells to bleomycin-treated animals to evaluate their antifibrotic and/or alveolar regenerative efficacy [175,176,177,178,179,180,181,182]. Previous studies have demonstrated that the administration of pluripotent- induced stem cells (iPSCs) to mice 24 h after a bleomycin challenge reduced the development of lung fibrosis [175,176]. The optimization of methods to successfully differentiate human embryonic stem cells (hESCs) [180] and pluripotent stem cells [177,178,181] into alveolar epithelial cells has allowed for the use of these derived cells for cell-based approaches. A mixture of epithelial cells (AT2, AT1 and Club cells) derived from hESCs and administered 7 days after bleomycin instillation showed a reversion in the fibrotic phenotype [180]. The authors concluded that engrafted cells may reduce fibrosis either by directly replacing the fibrotic tissue or indirectly by paracrine secretion of antifibrotic factors [180]. Improved approaches include the use of AT2 cells derived from iPSCs-induced pluripotent stem cells (iPSCs). The administration of mouse iPSC-derived AT2 cells 24 h after bleomycin instillation reduced the extent of the fibrosis since the collagen content was decreased and the lung tissue structure recovered [179]. Importantly, differentiated iPSCs labeled with a PKH26 cell tracker that were engrafted into the lungs and gave rise to AT2 and AT1 cells, recovering the cell numbers observed in control mice. Furthermore, inflammatory cell infiltration, as well as TNFα and IL-6 were reduced [179]. Interestingly, the transplantation of human iPSC-AT2 cells 15 days after bleomycin instillation, when fibrosis was already present in the lungs, induced a reduction in the collagen deposition by the inhibition of both TGF-β and α-SMA expression, and no evidence of inflammation or epithelial damage was observed in the transplanted lungs [182]. Consistently, fewer αSMA+ myofibroblasts developed, and fewer and smaller fibroblast foci were observed [182]. Of note, engraftment of these cells was not detected, so a paracrine effect is suggested to explain this. These findings reveal a potential use of iPSC-AT2 cells to resolve fibrotic damage. However, although this study demonstrates that the administration of iPSC-AT2 cells at the fibrotic stage of bleomycin-induced injury halts and reverses fibrosis, a more appropriate model of established fibrosis is necessary to test the actual potential of these cells to be used as a therapy for the treatment of IPF. 

The use of AT2 cells derived from iPSCs is, undoubtedly, an easy and efficient approach for the lung cell-based therapy that allows for autologous cell transplantation. However, although cells derived from iPSCs bypass the ethical concern associated with the use of human embryonic stem cells, it is necessary to establish protocols to ensure their correct administration and engraftment to guarantee their beneficial effects and to prevent dangerous side effects including the formation of tumors [183].

### 3.2. Clinical Human Studies

Epithelial and mesenchymal stem cells have been used in different clinical trials. The administration of human AT2 cells to IPF patients was demonstrated to be safe, well tolerated and showed no relevant side effects in patients with moderate and progressive IPF, during the supervised 12-month period [142]. However, it has not been assessed yet, whether this treatment provides an improvement in lung structure and functionality [142].

Interestingly, basal cells (BCs) have also been tested as a potential cell-based strategy for lung diseases. BCs act as airway stem cells that can give rise to all the epithelial cell types present in the mouse trachea and human airway epithelium [184,185]. Of note, they have shown to be able to migrate to lower parts of the respiratory tree and contribute to alveolar regeneration in cases of severe injury [101,106]. In this regard, a rare subset of human airway BCs characterized by the expression of SOX9 and with the capacity to regenerate functional alveoli has been identified [186,187]. Interestingly, the transplantation of autologous adult human SOX9+ airway BCs into bronchiectasis patients improved their lung function and no aberrant cell growth nor other related adverse events were observed during the whole follow-up period [186,187]. Hence, the use of airway basal cells in stem cell therapy for lung diseases may provide new opportunities to regenerate damaged lung tissue. Despite this strategy having not being tried for IPF, p63+ progenitors may represent an additional source that may give rise to alveolar lineages. Yet, to be used therapeutically, hypoxia, Notch and Wnt signaling should be regulated to avoid these cells from forming metaplastic structures with Krt5+ cells [65,77].

MSCs have been more extensively used. In human IPF, Averyanov and colleagues evaluated the safety, tolerability and efficiency of high cumulative doses of MSCs in fibrotic lungs and observed a rapid progressive course of severe to moderate IPF without significant adverse effects or differences in mortality after the administration of MSCs [188]. Another phase one trial study confirmed the safety of MSC (specifically autologous adipose-derived stromal cells (ADSCs)-stromal vascular fraction (SVF)) application, and reported improvements in life quality parameters and promising progression-free survival rates up to 24 months in 14 IPF patients, concluding that further and more complex clinical trials are needed to decipher the role of ADSCs in IPF pathogenesis and treatment [189,190]. Further phase one trials in nine patients with mild or moderate IPF and eight patients with moderately severe IPF obtained similar results [191,192]. Although a follow-up period of six months did not show improvements in lung function parameters and CT scores [191], the follow up 48 weeks post-transplantation revealed hints of therapeutic improvements with slower progression of fibrosis scores measured by CT scans and a slower decrease in the lung diffusion capacity for carbon monoxide in those patients receiving the higher MSC dosage [193]. Results about the potential of MSC administration as a preventive or curative therapy are heterogeneous, and in some cases contradictory due to differences in the specific cell type, dosage and the route of administration [194,195,196]. Future research efforts need to be addressed towards establishing the optimal conditions and defining the therapeutic window wherein treatment with these cells may be beneficial.

**Table 2 cells-11-02595-t002:** Cell therapy in preclinical mouse studies and clinical human studies.

Type of Study	Cell Source	Cell Delivery Route, Dose and Time of Administration	Time of Readouts and Results	Ref
**Preclinical mouse studies**	AT2 cells	Intratracheal route. A dose of 2.5 × 10^6^ cells/rat 14 days after a single intratracheal bleomycin administration	The animals were euthanized 21 days after bleomycin challenge. Treated rats after bleomycin instillation showed a reduction in the degree of fibrosis and a complete recovery to normal levels of surfactant proteins	[145]
	AT2 cells	Intratracheal route. A dose of 2.5 × 10^6^ cells/rat 3, 7 or 15 days after a single intratracheal bleomycin administration	The animals were euthanized 21 days after bleomycin challenge. Treated rats after bleomycin instillation showed reduced collagen deposition and reduction in the severity of pulmonary fibrosis (regardless the time point of AT2 cell treatment)	[146]
	AT2 cells	Intratracheal route. A dose of 2.5 × 10^6^ cells/rat 3 or 7 days after a single intratracheal bleomycin administration	The animals were euthanized 7 or 14 days after bleomycin challenge. Treated rats 7 days after bleomycin instillation showed an improvement in lung performance, structure and surfactant ultrastructure in bleomycin-induced lung fibrosis, while those treated 3 days after bleomycin instillation were only able to slightly recover the volume of AT2 and volume fraction of lamellar bodies in AT2	[147]
	Adult lung spheroid cells (LSCs)	Intravenous route. A dose of either 5 × 10^6^ syngeneic or allogeneic LSCs/rat 24 h after a single intratracheal bleomycin administration	The animals were euthanized 14 days after bleomycin challenge. Treated rats with allogeneic/syngeneic LSCs show an attenuation in the progression and severity of pulmonary fibrosis, decreasing apoptosis, protecting alveolar structures and increasing angiogenesis. Safety and efficacy of allogeneic LSCs treatment is demonstrated	[148]
	Human BM-MSCs	Intravenous route. A dose of 5 × 10⁵ cells/humanized mouse 2 days after a single intratracheal bleomycin administration	The animals were euthanized 7 or 21 days after bleomycin challenge. Treated humanized mice with human MSCs showed an attenuation of pulmonary fibrosis development. MSCs are suggested to suppress T-cell overactivation via PD-1 and PD-L1 interaction. Human MSCs have a therapeutic effect only in the early phase of pulmonary fibrosis	[151]
	Human BM-MSCs	Intravenous route. A dose of 0.5 × 10^6^ modified * or nonmodified cells/mouse 7 days after a single intratracheal bleomycin administration. * Cell modification refers to their prior transduction of miRNAs (let-7d or miR-154) using lentiviral vectors	The animals were euthanized 14 days after bleomycin challenge. Treated mice with human modified (let-7d) MSCs revealed shifts in animal weight loss, collagen activity after treatment and decrease in CD45+ cells, partially reducing the effects of bleomycin-induced lung injury. This study suggests the use of miRNA-modified BM-MSCs as a potential therapeutic strategy	[152]
	BM-MSCs	Intravenous route. A dose of 5 × 10⁵ cells/mouse immediately after or 7 days after a single intratracheal bleomycin administration	The animals were euthanized 14 days after bleomycin challenge. Immediately after bleomycin instillation, treated mice showed an amelioration in the fibrotic injuries, while those treated 7 days after bleomycin instillation, even though engraftment was not inhibited, the ability of the cells to alter the course of disease progression was eliminated	[155]
	BM-MSCs	Intravenous route. A dose of 2.5 × 10^6^ cells/rat immediately after or 7 days after a single intratracheal bleomycin administration	The animals were euthanized 7, 14 or 28 days after bleomycin challenge. The present study demonstrates that when MSCs were administered after bleomycin challenge, exogenous MSCs were immediately detected in lung tissues from rats sacrificed at different time points and the number of MSCs in the lung tissue increased over time, while this did not happen to the group treated after 7 days of bleomycin instillation	[156]
	BM-MSCs	Intravenous route. Two doses of 0.5 × 10^6^ cells/mouse. The first one was administered after a single oropharyngeal bleomycin administration and the second dose, 3 days after the first dose	The animals were euthanized 14 days after bleomycin challenge. This study demonstrates that BM-MSCs expressing keratinocyte growth factor via an inducible lentivirus protects against bleomycin-induced lung fibrosis	[157]
	Human BM-MSCs	Intravenous route. A dose of 5 × 10⁵/mouse 24 h after a single intratracheal bleomycin administration	The animals were euthanized 14 days after bleomycin challenge. In this study, the authors show that MSCs can correct the inadequate communication between epithelial and mesenchymal cells through STC1 (Stanniocalcin-1) secretion after bleomycin instillation	[158]
	BM-MSCs	Intratracheal route. A dose of either 5 × 10⁵ hypoxia-preconditioned or control cells/mouse 3 days after a single intratracheal bleomycin administration	The animals were euthanized 7 or 21 days after bleomycin challenge. This study reports that hypoxia-preconditioned BM-MSCs improve pulmonary functions and reduce inflammatory and fibrotic mediators after bleomycin-induced lung fibrosis	[159]
	Oncostatin M (OSM)-preconditioned BM-MSCs	Intratracheal route. A dose of either 2 × 10⁵ oncostatin M (OSM)-preconditioned or control cells/mouse 3 days after a single intratracheal bleomycin administration	The animals were euthanized 7 or 21 days after bleomycin challenge. Transplantation of OSM-preconditioned MSCs significantly improved pulmonary respiratory functions and downregulated expression of inflammatory factors and fibrotic factors after bleomycin instillation	[160]
	BM-MSCs	Intravenous route. A dose of 1 × 10^6^ cells/mL/rat 14 days after a single intratracheal bleomycin administration	The animals were euthanized 28 days after bleomycin challenge. Animals treated with BM-MSCs showed a significant decrease in the alveolar wall thickening, in the inflammatory infiltrate and in the collagen fiber deposition. The conclusion of the study was that the therapeutic pulmonary anti-fibrotic activity of BM-MSCs is mediated through their anti-inflammatory properties and inhibition of SMAD-3/TGFβ expression	[161]
	Resident lung MSCs (luMSCs)	Intravenous route. A dose of either 0.15 × 10^6^ or 0.25 × 10^6^ cells/mouse immediately after a single intratracheal bleomycin administration	The animals were euthanized 14 or 35 days after bleomycin challenge. Treated animals showed a decrease in numbers of lymphocytes and granulocytes in bronchoalveolar fluid and display reduced collagen deposition. Also, treatment with luMSCs significantly decreased weight loss associated with bleomycin and increased survival from 50% at 14 days with bleomycin alone to 80% when mice had been treated with luMSCs	[162]
	BM-MSCs	Intravenous route. A dose of 5 × 10⁴ allogeneic cells/g/mouse 6–8 h or 9 days after a single intranasal bleomycin administration	The animals were euthanized 28 days after bleomycin challenge. Early treatment with allogeneic MSCs protected the lung architecture and significantly reduced fibrosis, apoptosis and IL1-production, while delayed MSC treatment failed to protect the mice from bleomycin-induced lung fibrosis. Of note, this is the first study to definitively show the importance of naturally derived HFG in MSC protection in the bleomycin model	[164]
	Amnion-MSCs vs. BM-MSCs vs. human amniotic epithelial cells (hAECs)	Intravenous route. A dose of 1 × 10^6^ cells/mouse 3 days after introducing the second bleomycin injury (bleomycin administration was done intra-nasally, and the second dose was given 7 days after the first one)	The animals were euthanized 17 or 31 days after bleomycin challenge. This study concluded that amnion-MSCs may be more effective than BM-MSCs and hAECs in reducing injury following delayed injection in the setting of repeated lung injury	[165]
	ADSCs	Intravenous route. A dose of either 5 × 10⁵ young-donor or old-donor cells/mouse 24 h after a single intratracheal bleomycin administration	The animals were euthanized 21 days after bleomycin challenge. Treated old mice with young ADSC displayed a greater reduction in fibrosis, oxidative stress, MMP-2 activity and apoptosis markers than mice treated with old ADSCs	[168]
	ADSCs	Intravenous route. A dose of either 2.5 × 10⁴ or 2.5 × 10⁵ cells/mouse immediately after subcutaneous bleomycin administration for 7 days	The animals were euthanized 7 or 21 days after bleomycin challenge. ADCSs accumulated in the pulmonary interstitium and inhibited both inflammation and fibrosis in the lung. Treated mice showed decreased lung fibrosis and inflammation in a dose-dependent manner	[169]
	ADSCs (human)	Intraperitoneal route. During the latter 2 months of bleomycin exposure * 3 × 10⁵ human cells were administered repeatedly at the same time as bleomycin. * Bleomycin was injected intratracheally in eight biweekly doses	The animals were euthanized 14 days after bleomycin challenge. Treated mice showed decreased lung fibrosis, inflammatory cell infiltration, epithelial hyperplasia, TGFβ expression and epithelial apoptosis	[170]
	ADSCs	Intravenous route. A dose of 5 × 10⁵ cells/mouse 24 h after a single intratracheal bleomycin administration	Mice treated with ADSCs showed attenuated bleomycin-induced lung and skin fibrosis and accelerated wound healing. This study suggests that ADSCs may prime injured tissues and prevent end-organ fibrosis	[171]
	ADSCs (human)	Intravenous route. A dose of 40 × 10^6^/kg body weight/mouse 3, 6 and 9 days after a single intratracheal bleomycin administration	The animals were euthanized 24 days after bleomycin challenge. Mice treated with ADSCs showed a higher increase in survivability, organ weight reduction and collagen deposition when compared to those treated with pirfenidone. Also, ADSCs potently suppressed profibrotic genes induced by bleomycin and also inhibited pro-inflammatory related transcripts	[172]
	Human Placental MSCs of fetal origins (hfPMSCs)	Intravenous route. A dose of 1 × 10⁵ cells/mouse 3 days after a single intratracheal bleomycin administration	The animals were euthanized 0, 7 and 28 days after bleomycin challenge. Treatment with hfPMSCs showed that these cells can attenuate bleomycin-induced lung inflammation and fibrosis in mice, in part through a mechanism by attenuating MyD88-mediated inflammation	[173]
	Amniotic fluid stem cells (AFSCs)	Intravenous route. A dose of 1 × 10^6^ cells/mouse either 2 h or 14 days after a single intratracheal bleomycin administration	The animals were euthanized 3, 14, 28 days after bleomycin challenge, depending on the group. Treated mice at both time points showed inhibition in the changes in lung function associated with bleomycin-induced lung injury and decreased collagen deposition	[174]
	iPSCs	Intravenous route. A dose of 2 × 10^6^ cells/mouse 24 h after a single intratracheal bleomycin administration	The animals were euthanized 21 days after bleomycin challenge. Treated mice after bleomycin showed an inhibition of EMT, inflammatory response and TGF-β1/Smad2/3 signaling pathway	[175]
	iPSCs	Intravenous route. A dose of 2 × 10^6^ cells/mouse (cells either lacking c-Myc or in condition medium) 24 h after a single intratracheal bleomycin administration	The animals were euthanized 3, 7, 14 or 21 days after bleomycin challenge. Treated mice, after bleomycin instillation, showed an attenuation in collagen content, diminished neutrophil accumulation and rescued pulmonary function and recipient survival after bleomycin-induced lung injury	[176]
	Mouse iPSCs-derived AT2 cells	Intravenous route. A dose of 5 × 10⁵ cells/mouse 24 h after a single intratracheal bleomycin administration	The animals were euthanized 13 days after bleomycin challenge. Treated mice after bleomycin have decreased collagen deposition and lung inflammation	[179]
	Human iPSCs-derived AT2 cells	Intratracheal route. A dose of 3 × 10^6^ cells/rat 15 days after a single intratracheal bleomycin administration	The animals were sacrificed 21 days after bleomycin administration. Transplanted lungs showed no inflammation, no edema, no epithelial damage and reduced fibrosis	[182]
	AT2, AT1 and Club cells derived from human embryonic stem cells (hESCs)	Intratracheal route. A dose of 1 × 10⁵ differentiated hESCs/mouse 7 days after a single intratracheal bleomycin administration and immediately after sublethal irradiation to avoid graft rejection	The animals were euthanized 14 days after bleomycin challenge. Treated mice, after bleomycin instillation, showed an increase in progenitor number in the airways and reduced collagen content	[180]
**Clinical human studies**	AT2 cells (heterologous)	Intratracheal route. A total of 16 IPF patients. Four doses of 1000–1.200 × 10^6^ cells/patient	Enrolled patients were monitored for 1 year. Administered AT2 cells were both safe and well tolerated. There was no deterioration in pulmonary function, respiratory symptoms or disease extent after 12 months of follow-up. This study lacks a control group due to ethical issues	[142]
	SOX9 + BCs (autologous)	Endobronchial route. A total of 2 bronchiectasis patients. A dose of 1 × 10^6^ cells/kg body weight/patient	This study was the first autologous SOX9 + BCs transplantation clinical trial. Lung tissue repair and pulmonary function enhancement was observed in patients 3–12 months after cell transplantation	[186]
	SOX9 + BCs (autologous)	Endobronchial route. A total of 7 bronchiectasis. A dose of 1 × 10^6^ cells/kg body weight/patient	Enrolled patients were monitored for 1 year. Transplantation of autologous SOX9 + BCs had positive effects and is safe for patients with bronchiectasis	[187]
	BM-MSCs (allogeneic)	Intravenous route. A total of 20 patients with usual interstitial pneumonia and a history of lung function decline over the last 12 months, among other characteristics. Two doses of 200 × 10^6^ cells/patient, every 3 months	Enrolled patients were monitored for 1 year. This study concluded that therapy with high doses of allogeneic MSCs is a safe and promising method to reduce disease progression in IPF patients with rapid pulmonary function decline	[188]
	ADSC-SVF (stromal vascular function)	Endobronchial route. A total of 14 IPF patients. A dose of 0.5 × 10^6^ cells/kg body weight/patient/month (a total of 3 months)	Enrolled patients were monitored for 1 year. There was no formation of ectopic tissues and no difference in adverse events compared to placebo effect. Treatment was safe for IPF patients	[189]
	ADSC-SVF (stromal vascular function)	Endobronchial route. A total of 14 IPF patients. A dose of 0.5 × 10^6^ cells/kg body weight/patient/month (a total of 3 months)	This study is the follow-up of the study above. They saw a significant functional decline was observed at 24 months after the first administration and highlighted the need of further clinical trials using these cells	[190]
	BM-MSCs (allogeneic)	Intravenous route. A total of 9 IPF patients. A dose of either 20 × 10^6^, 100 × 10^6^ or 200 × 10^6^ cells/patient	Safety was assessed for 15 months in total. No treatment-emergent serious adverse events were reported in this study. This trial (called AETHER) was the first clinical trial conducted for 15 months to assess the safety of a single intravenous infusion of BM-MSCs	[191]
	Placental MSCs (allogeneic)	Intravenous route. A total of 8 IPF patients. A dose of either 1 × 10^6^ or 2 × 10^6^ cells/kg body weight/patient	Enrolled patients were followed for 6 months. Intravenous administration of these cells was proven to be feasible and to have a good short-term safety profile in patients with moderately severe IPF	[192]
	BM-MSCs (allogeneic)	Intravenous route. A total of 9 IPF patients. A dose of either 20 × 10^6^, 100 × 10^6^ or 200 × 10^6^ cells/patient	This study is a follow-up of the AETHER trial. The subjects receiving the higher dose demonstrated better results when compared to those receiving the lowest dose	[193]

## 4. Current Challenges

Idiopathic pulmonary fibrosis is a rare and fatal disease with no cure. Unfortunately, there are still a lot of unanswered questions about its pathophysiology and the mechanisms underlying its initiation and progression, and little progress would be possible without proper models to study IPF. In vivo models rely on measuring the response to injury caused by bleomycin, a tumor chemotherapeutic agent that has been used to study pathological fibrosis in the lung for around 50 years [197]. The lack of treatment options and the heterogeneity in the natural progression of the disease among patients highlights the importance of developing new strategies, not only to tackle IPF emergence and progression, but also to try to reverse fibrosis and restore functional tissues. The ideal therapeutic strategy should first, eliminate the source of the injury, although in most cases this is unknown. Next, it should remove the fibrotic tissue that is precluding an efficient gas exchange and physiological cellular crosstalk within alveoli; and third, it should promote the regeneration of damaged tissues to restore tissue function and recover a homeostatic state. In this regard, cell therapy offers great opportunities towards disease healing. Nonetheless, we must take into account that most of the cell replacement therapy strategies have been assayed as a pretreatment in preclinical studies, being administered in the inflammatory stage (24 h-7 days after bleomycin challenge) rather than in the fibrotic phase which impedes an assessment of the efficacy of the transplantation of the different cell types in resolving fibrosis [198,199]. In general, cell transplantation during these initial stages showed promising results since the progenitor cells that were administered inhibited inflammation and impeded fibrogenesis. However, controversial results are found when cells were administered during the fibrotic phase, raising questions about their actual therapeutic utility [198,199]. These issues most likely arose due to the injury model used, since the single-dose bleomycin model is an acute injury model that resolves naturally and therefore, cannot be compared to the chronic and progressive nature of human IPF. As aforementioned, a better model of established fibrosis needs to be used to analyze the therapeutic potential of transplanted cells in halting and/or reversing fibrosis.

All in all, there are still limitations in the use of stem cells for therapy, for example, the ability of exogenous stem cells to home into the desired tissue has been shown to be relatively low. Currently, the best strategy to promote tissue regeneration seems to be to promote endogenous cells to regenerate lost or damaged tissue. However, there is still much work ahead to unravel how these cells achieve regeneration successfully and which cells do it best.

## 5. Future Perspectives and Conclusions

The remarkable plasticity that airway and alveolar epithelial cells exhibit provides the lung with a wide variety of region-specific stem and progenitor cell populations that participate in regeneration after injury. Leveraging the mechanisms governing lung regeneration may help to develop strategies aimed towards inducing tissue repair in the diseased lung.

Of note, just as important as identifying the reparative cells that can contribute to the process of lung regeneration is the understanding of the microenvironment that dictates the path of an endogenous or exogenous progenitor. Understanding the interactions between epithelial progenitors and their niche and how these signaling inputs are integrated is crucial to enhance inefficient mechanisms of repair in areas of advanced fibrosis where scar tissue impedes the survival of healthy cells.

Over many years, replacing pathologic epithelial cells with exogenous progenitors has been thought to be a promising treatment for fibrotic pulmonary diseases. However, the potential of stem cell-based therapy is still not a reality despite significant advances accomplished in the field. Currently, a better knowledge about which cells can execute regenerative responses as well as the development of optimized methods for expanding lung progenitors encourage the clinical application of these therapies. Yet, more mechanistic research is needed in order to demonstrate the actual potential of epithelial progenitor transplantation in patients with lung fibrosis as well as the long-term fate of these cells, including the possibility of their contribution to the development of tumors.

Research towards the study of the mechanisms of quiescence to return to steady-state conditions with the active signals and interactions that maintain tissue homeostasis is of paramount importance in order to restore tissue balance after the therapeutic induction of lung regeneration when tissue repair has been achieved.

In addition to the administration of exogenous cells, the stimulation of endogenous progenitors has also been proposed as a therapeutic approach to promote lung regeneration. Lessons from lung development and postnatal alveologenesis may provide clues to developing this strategy. However, other limitations and unanswered questions for the successful achievement of the therapeutic induction of lung regeneration exist. First, it is unknown whether endogenous AT2 cells in fibrotic lungs are able to accomplish successful lung regeneration. Many questions arise because of this; how damaged are they? Do they retain the capacity to proliferate and initiate differentiation into functional AT1 cells? Are they going to find the way to initiate self-organization of functional alveoli? If this process is instructed by ECM that is highly altered in IPF, is it going to be possible to regenerate functional alveolar units with an adequate cell distribution in this context? Should the microenvironment be restored first, so signals received by progenitors are the appropriate ones necessary for regeneration? 

A better understanding of human lung regeneration is required. The development of new models is particularly necessary to investigate the mechanisms involved in repair and regeneration and to find out how to promote lung regeneration to treat human lung disease. Larger animal models, such as those involving non-human primates, ferrets and pigs may be necessary for this purpose as their respiratory anatomy closely resembles the anatomy of the human respiratory system. Alternatively, precision cut lung slides (PCLS) from healthy and IPF lungs allow for the study of human lung progenitors and regeneration in the human lungs. These models promise to close the gap between basic research and clinical application, providing the basis for a deeper understanding of lung physiology and pathology in humans and allowing the design of more curative therapies. 

Much research is still needed in order to develop successful strategies focused on cell-based therapies to promote lung regeneration in combination with other therapeutic approaches, targeting pathological events and addressed towards eliminating aberrant cells and the matrix to restore lung architecture and regenerate functional alveolar units for breathing.

## Figures and Tables

**Figure 1 cells-11-02595-f001:**
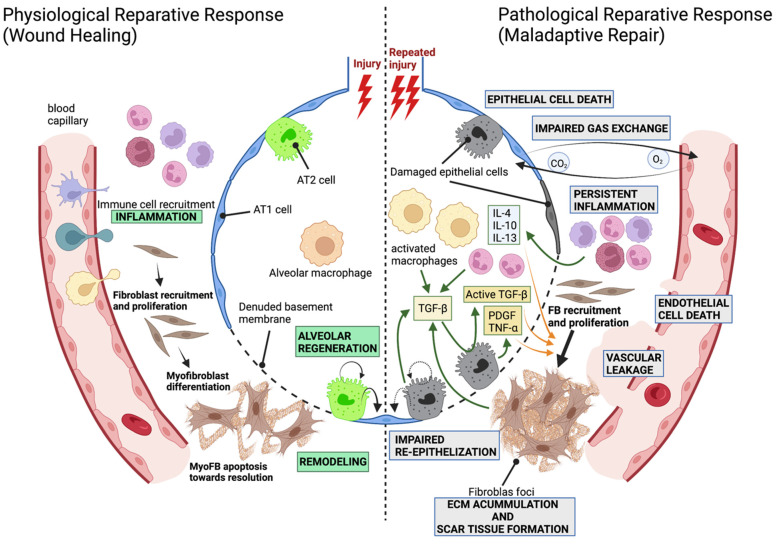
**Alveolus under normal and aberrant wound healing.** Left: In normal circumstances, after injury in the alveolar epithelium, immune cells are recruited into the lung interstitium and an acute inflammatory response takes place. The cytokines released during inflammation by immune cells together with other factors produced by epithelial and endothelial cells elicit fibroblast proliferation and myofibroblast differentiation. Myofibroblasts (MyoFB) secrete extracellular matrix (ECM) that contributes to tissue remodeling and alveolar type 2 (AT2) cells, that serve as alveolar stem cells in the adult lung, self-renew and give rise to AT1 cells to regenerate the alveolar epithelium. Then, MyoFB are eliminated by apoptosis and homeostasis is restored. Right: Upon repetitive injury, normal wound healing turns into an aberrant response where damaged epithelial cells lose their ability to properly repopulate the alveolar epithelium, myofibroblasts in the fibroblast foci produce excessive amounts of ECM that results in tissue scarring, and vascular leakage takes place after endothelial cell death. As a consequence, gas exchange is impaired. The non-resolved tissue repair includes persistence of inflammation with interstitial immune cell accumulation and alveolar immune cell infiltrates. The release of anti-inflammatory citokines such as IL-4, IL-10 and IL-13 impacts fibroblast recruitment and proliferation and promotes macrophage polarization into alternatively activated macrophages. These macrophages, neutrophils, myofibroblasts and AT2 cells produce TGF- β, a major player involved in this process, that together with platelet-derived growth factor (PDGF) and tumor necrosis factor alfa (TNF-α) produced by AT2 cells, contribute to fibroblast recruitment, proliferation and differentiation. This creates a fibrotic loop that contributes to disease progression in IPF (Image created with BioRender).

**Figure 2 cells-11-02595-f002:**
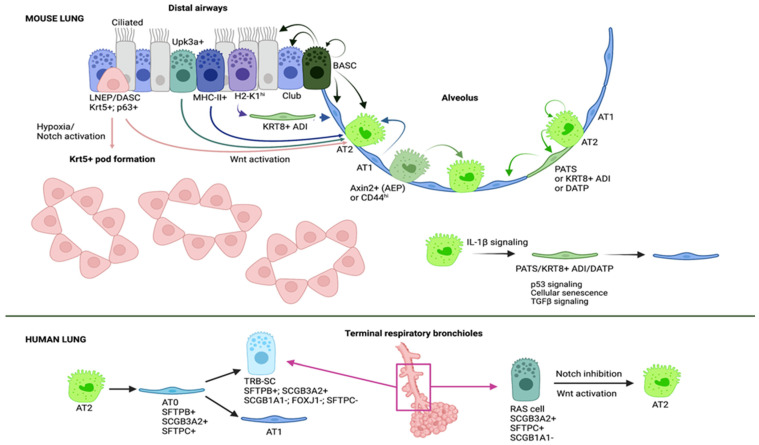
**Distal airway/alveolar progenitors contributing to lung regeneration.** A myriad of cells serve as alveolar progenitors upon damage. AT2 cells can proliferate and differentiate into AT1 cells after going through an intermediate cell state named pre-alveolar type-1 transitional cell state (PATS), keratin 8-positive alveolar differentiation intermediate (Krt8+ADI) or damage associated transient progenitors (DATPs). IL-1β produced by interstitial macrophages promotes AT2 differentiation into this intermediate cell type that displays a transcriptional signature of p53 signaling, cellular senescence and TGFβ signaling. Alveolar epithelial progenitors (AEPs) represent a subset of AT2 cells characterized by the expression of Axin2 that act as the principal progenitor cell population during injury-induced alveolar regeneration. CD44^hi^-expressing AT2 cells show an increased proliferative capacity also contributing to the regeneration of the alveolar epithelium. Interestingly although rarely, AT1 cells are able to dedifferentiate and give rise to AT2 cells. Bronchoalveolar stem cells (BASCs) are cells contributing to both alveolar and airway regeneration because of their ability to self-renew and to give rise to AT2, AT1, club and ciliated cells. Furthermore, subsets of club cells such as Upk3a+ subset, H2-K1^hi^ subset and MHC-II+ subset can differentiate into AT2 cells; the latter go through a transitional state similar to KRT8+ ADI cells to give rise to AT1 cells. Of note, H2-K1^hi^ and MHC-II+ club cells show an identical transcriptional signature suggesting that they are the same subpopulation. A rare population of p63+ cells in terminal bronchioles have shown the ability to activate Krt5 expression and expand and migrate to sites of injury. There, these cells give rise to AT2 cells or form pod-like metaplastic structures in a process regulated by hypoxic conditions, Notch signaling and Wnt signaling. Studies in human and non-human primate models have identified two interesting cell populations: AT0 cells, a novel bi-potential transient state that arises in the differentiation from AT2 cells into either terminal respiratory bronchiole secretory cells (TRB-SCs) or AT1 cells; and RAS cells, an airway secretory cell population that can differentiate into AT2 cells in a process regulated by Notch and Wnt signaling (Image created with BioRender).

**Figure 3 cells-11-02595-f003:**
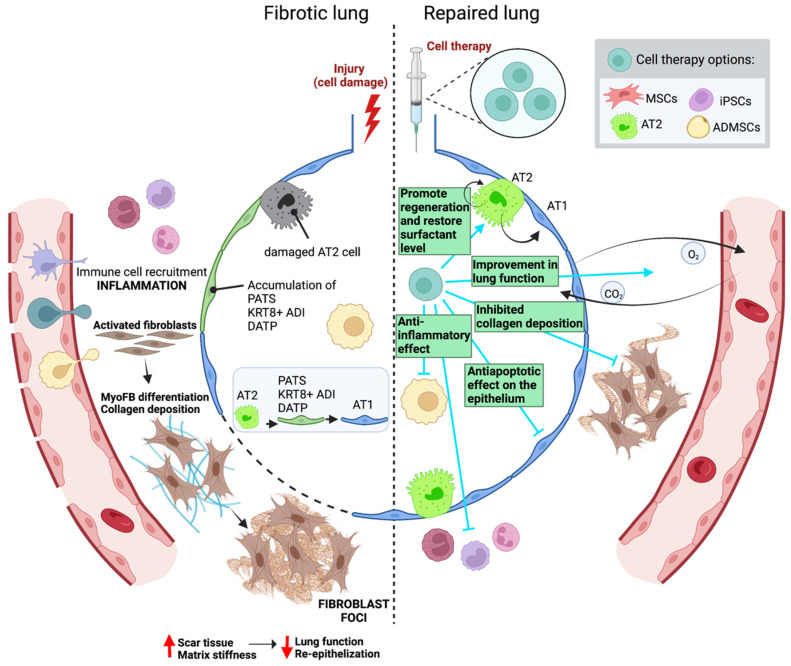
**Stem cell-based therapies in lung fibrosis.** Left: In the fibrotic lung the excess of collagen deposition evokes tissue scarring increasing matrix stiffness which impairs lung function. Re-epithelization is also affected due to alveolar type 2 (AT2) cells’ failure to completely differentiate into AT1 cells as they are stuck in an intermediate cell state known as damage-associated transient progenitors (DATPs), pre-alveolar type-1 transitional cell state (PATS) or keratin 8-positive alveolar differentiation intermediate (Krt8+ADI). Right: Adult mesenchymal stromal/stem cells (MSCs), induced pluripotent stem cells (iPSCs), AT2 cells and adipose mesenchymal stem cells (ADMSCs) have been proposed for cell transplantation to induce alveolar regeneration. Stem cell-based therapy has proven to exert beneficial effects by promoting epithelial regeneration, restoring surfactant levels, inhibiting collagen deposition, exerting an antiapoptotic effect on the epithelium and also showing anti-inflammatory effects; all of this contributes to improved lung function (Image created with BioRender).

**Table 1 cells-11-02595-t001:** Murine models to study lung fibrosis in vivo.

Murine Models	Main Pathological Features	Pros	Cons	Ref
Bleomycin (single dose (I.T, I.N, I.V) or repeated doses (I.T, I.N, I.P, O.A, I.V))	Epithelial cell injury. Fibroblast foci. Macrophage oxidative stress. Fiber deposition	Some of the molecular signatures as well as some histopathological hallmarks at distinct stages of bleomycin-induced lung fibrosis resemble those encountered in human fibrotic lung diseases. Quick development of fibrosis. Relative ease of induction, reproducibility and versatility. Economical	Important role of inflammation in the development of fibrosis. Some reports show that the fibrotic lesions resolved naturally after day 21–28, while other recent studies indicated persistence of fibrosis, albeit with less inflammation as long as 6 months after a single or repetitive bleomycin treatment(s). However, the chronic model that uses several doses of bleomycin may overcome the natural-resolving fibrosis handicap	[45,47,48,49,50,51,52,53,54]
Silica	Fibrotic nodules develop around silica deposits and silica fibers are easily identified both by histology and polarization microscopy. Macrophage NALP3 inflammasome activation regulates disease development	Development of fibrotic nodules that resemble lesions that develop in humans following exposure to mineral fibers and particulate aerosols. Persistence of fibrotic lesions due to diminished clearance of silica particles from the lungs	Highly expensive and difficult delivery, prolonged waiting periods until fibrosis develops (4–16 weeks), lack of reproducibility of fibrotic pattern, absence of usual interstitial pneumonia (UIP)-like lesions	[48,55]
Asbestosis	Asbestos bodies embedded within the fibrous tissue, fewer myofibroblasts foci and bronchial wall fibrosis. In some cases, the pattern of UIP can be also present	Recapitulates asbestos exposure in human lung fibrosis	A single intratracheal administration elicits an uneven distribution of fibrosis between lungs which also tends to develop in the core of the lung rather than in the subpleura. The fibrosis developed from the inhalation model is more peripheral but requires at least a month for fibrosis to develop	[55]
Hyperoxia	Hypoalveolarization. Increased elastin and collagen-I deposition by α-actin-positive myofibroblasts. Increased periostin expression in the alveolar walls, particularly in areas of interstitial thickening	Allows the study of prolonged exposure to supplemental oxygen	Additional studies investigating controversial molecular mechanisms underlying hyperoxia-induced cell injury should be performed since these may be helpful in future pharmaceutical interventions	[56,57]
Acid instillation	Pattern of fibrosis involves interstitial rather than alveolar consolidation	Allows studies of hypoxemia, permeability injuries and effects of hyperoxia. It also models fibroproliferative changes seen with ALI and ARDS	Modifications (e.g., a fluid bolus, supplemental oxygen and careful monitoring to be assured of surviving the procedure) are imperative because without them the animals die of lung injury before the development of lung scarring	[48]
Cytokine overexpression	Epithelial apoptosis and myofibroblast accumulation. Airway and parenchymal fibrotic response	Ability to dissect downstream signaling events relevant to specific fibrotic-inducing cytokines. Fibrotic scarring tends to be more persistent in some models than those produced by bleomycin	Models limited to dissecting specific pathways. Highly variable and heterogeneous kinetics of injury regarding severity, lesions extension and lack of reproducibility	[55]
Fluorescent isothiocyanate (FITC)	AEC injury. Vascular leak	Relatively reproducible and persistent fibrotic phenotypes. Easily trackable fluorescence-labeled fibrotic tissues	Lack representative UIP and inflammatory infiltrates preceding fibrosis. Technical issues regarding FITC particles may compromise model robustness. Limited human relevance since this type of injurious stimulus has never been described in humans	[48,55]
Radiation-induced	AEC injury. Vascular remodeling. MSCs regulate repair responses	Results in fibrosis and can be local or systemic if other organs are not shielded	Fibrosis takes a long time to develop. Mainly dependent on inflammation and free-radical-mediated DNA damage and less on TFG-B	[48,55]
Familial models	Depends on the altered gene of study	Useful to study the disease genetic background	Mutations may produce a susceptible phenotype, requiring also a second hit from environmental origin to partially recapitulate the human phenotype	[55]
Humanized (NOD/SCID mice)	Immunodeficient mice	It allows for cell trafficking during different stages of fibrosis development and progression, offers insights into role of different fibroblast populations and dissects the contribution of epithelial-fibroblast crosstalk in the absence of immune cells	May not be representative of human disease where immune cells play a role. High cost and requires specialized housing.	[55]

I.T.: intratracheal; I.N.: intranasal; I.P.: intraperitoneal; O.A.: oropharyngeal; I.V.: intravenous.

## Data Availability

Not applicable.

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
