# Peer review of "Promises and Challenges of Cell-Based Therapies to Promote Lung Regeneration in Idiopathic Pulmonary Fibrosis"

_cells, 2022, doi:10.3390/cells11162595_

Round 1

Reviewer 1 Report

The review paper of Egea-Zorrilla et al. is devoted to the cell-based regenerative therapy of Idiopathic Pulmonary Fibrosis (IPF).

The review is well written and in my opinion can be published after some polishing and addressing the issues listed below.

1.      Fig. 1 seems somewhat primitive. I would recommend supplementing it with the main pro- and antifibrotic cytokine cascades in normal regeneration and in the IPF pathogenesis (see the pathogenesis of ARDS in Fig.2 at the link as an example https://pubmed.ncbi.nlm.nih.gov/33016981/).

2.      The story about lung stem/progenitor cells as well as lung regeneration mechanisms could and needs also be illustrated.

3.      Fig. 2 looks a little bit better than previous, but its left side partially duplicates the right side of Fig.1. However, I would also add here the main regenerative factors of stem cell therapy in order to clarify a mechanism of action.

4.      Refs. 146-150 are duplicated as Ref. 202-206. I would recommend to double-check all the list.

5.      The Discussion section looks a bit strange in a review article. I would think about how to rearrange the content and phrase the name differently.

6.      The Conclusion section is too large and vague. Taking together with the previous point, a restructuring of the last two sections is firmly suggested and certainly required.

Reviewer 3 Report

1) Abstract: L 16-27. The lung epithelium is constantly exposed to harmful agents present in the air that we  breath making it highly susceptible to be damaged. However, in response to injury the lung exhibits  a remarkable capacity to regenerate injured tissue thanks to the presence of distinct stem and progenitor cell populations along the airway and alveolar epithelium. Mechanisms of repair are affected in chronic lung disease such as idiopathic pulmonary fibrosis (IPF), a progressive life-threatening disorder characterized by loss of alveolar structures, excessive deposition of extracellular matrix components causing the distortion of tissue architecture that limits lung function and impairs  tissue repair. Here, we review the most recent findings of epithelial cells with progenitor behavior  that contribute to tissue repair as well as the mechanisms involved in mouse and human lung regeneration. In addition, we describe therapeutic strategies to promote or induce lung regeneration  and the cell-based strategies tested in clinical trials for the treatment of IPF. Finally, we discuss the challenges, concerns and limitations of applying these therapies of cell transplantation in IPF patients. The abstract is quite rumbling please divide it in different sections (i.e. background, aim and conclusions).

2) Abstract. L 16-27. Please add a sentence regarding the conclusions of the paper.

3) Introduction. L 33-36.  Interstitial lung diseases (ILDs) is a group of more than 200 diverse parenchymal  pulmonary disorders mainly characterized by alveolar and interstitial inflammation  and/or fibrosis [1]. Idiopathic Pulmonary Fibrosis (IPF) is a chronic, progressive and fatal  disease of unknown etiology and it is considered one of the most representative types of  lung fibrosis within the ILDs [2]. Please improve this paragraph and add these recent references:

A- Regeneration or Repair? The Role of Alveolar Epithelial Cells in the Pathogenesis of Idiopathic Pulmonary Fibrosis (IPF). Cells. 2022 Jun 30;11(13):2095. doi: 10.3390/cells11132095. 

B- Epithelial-Mesenchymal Transition: A Major Pathogenic Driver in Idiopathic Pulmonary Fibrosis? Medicina (Kaunas). 2020 Nov 13;56(11):608. doi: 10.3390/medicina56110608.

4) Introduction. L 44. Several therapeutic interventions have been tested to  treat IPF but none have been proven to succeed. Please improve this paragraph and add these references:

a- Evolution and treatment of idiopathic pulmonary fibrosis. Presse Med. 2020 Jun;49(2):104025. doi: 10.1016/j.lpm.2020.104025. 

b- Pharmacological Interactions of Nintedanib and Pirfenidone in Patients with Idiopathic Pulmonary Fibrosis in Times of COVID-19 Pandemic. Pharmaceuticals (Basel). 2021 Aug 20;14(8):819. doi: 10.3390/ph14080819.

5) Introduction. L 65-70.  Although the etiology of IPF remains unknown it is now  generally accepted that alveolar epithelial cell death and/or dysfunction and the  consequent basement membrane denudation with loss of epithelial integrity play a key  role in the initiation and/or progression of lung fibrosis [18]. Yet, it is very unlikely that a  single cell type or event is responsible for IPF pathogenesis. Rather, the complex crosstalk  between the alveolar epithelium and the neighboring mesenchyme that is disrupted in  this context, may be crucial driving fibrogensis [5, 9]. Please improve this sentences and discuss the data reported in study (A) and (B) of point 3.

6) Introduction. L 65-70.  Although the etiology of IPF remains unknown it is now generally accepted that alveolar epithelial cell death and/or dysfunction and the consequent basement membrane denudation with loss of epithelial integrity play a key  role in the initiation and/or progression of lung fibrosis [18]. Yet, it is very unlikely that a single cell type or event is responsible for IPF pathogenesis. Rather, the complex crosstalk  between the alveolar epithelium and the neighboring mesenchyme that is disrupted in this context, may be crucial driving fibrogensis [5, 9]. Please add here a brief description of the study aims.

7) 4. Discussion L486-489.  Idiopathic pulmonary fibrosis is a rare and fatal disease with no cure. Unfortunately, there are still a lot of unanswered questions about their pathophysiology  and the mechanisms underlying its initiation and progression, and little progress would  be possible without proper models to study IPF. Please underline the novelty of the study

8) 5. Conclusions. L 553-556. Much research is still needed in order to develop successful strategies focused on  cell-based therapies to promote lung regeneration in combination with other therapeutic  approaches targeting pathological events and addressed towards eliminating aberrant  cells and matrix to restore lung architecture and regenerate functional alveolar units for  breathing. Please underline the clinical implication of the manuscript.

Reviewer 4 Report

I feel this review is very comprehensive, covering most of the existing knowledge and literature from IPF pathogenesis to cell therapy. I only have two minor suggestions:

1. In the 3. Cell therapy in lung fibrosis section, can the authors re-organize to have a separate section for "clinical human studies" in the text. This will make this part standing out, and also be consistent with the table 2.

2. If the authors put down 3.2.1 Adipose mesenchymal stem cells as a subtitle, should they put down Adult mesenchymal stromal/stem cells as a parallel one under 3.2?

3. For the last part, instead of using subtitle "Conclusion", it's more appropriate to use subtitle "Future perspective", where the authors did discuss about the future directions and limitations of cell therapy for IPF treatment.

Round 2

Reviewer 1 Report

The authors have revised the manuscript and addressed all my comments. I think the manuscript currently can be accepted for publication.